# HPV-Driven Head and Neck Cancer: The European Perspective

**DOI:** 10.3390/v17050662

**Published:** 2025-04-30

**Authors:** Wojciech Golusiński, Ewelina Golusińska-Kardach, Piotr Machczyński, Mateusz Szewczyk

**Affiliations:** 1Department of Head and Neck Surgery, University of Medical Sciences, 61-701 Poznan, Poland; wgolus@ump.edu.pl (W.G.); piotr.machczynski@wco.pl (P.M.); 2The Greater Poland Cancer Centre, 61-701 Poznan, Poland; 3Department and Clinic of Dental Surgery, Periodontal Diseases and Oral Mucosa, Poznan University of Medical Sciences, 60-812 Poznan, Poland; ekardach@ump.edu.pl

**Keywords:** HPV, oropharynx, epidemiology, Europe, head neck

## Abstract

Head and neck squamous-cell carcinoma (HNSCC) has long been associated with tobacco and alcohol use. In the last two decades, however, human papillomavirus (HPV) infection has emerged as an important driver of these cancers, particularly in the oropharynx. The eighth edition of the American Joint Committee on Cancer (AJCC) staging system now defines HPV+ and HPV− OPSCC as separate entities. Although our understanding of HPV+ HNSCC continues to improve, it can be challenging to keep up to date with the growing body of evidence. In this context, the present narrative review was carried out to provide an overview of HPV-driven head and neck cancer, with an emphasis on Europe. We review the latest evidence on epidemiology, diagnosis, and treatment, including recent trends towards treatment de-intensification and future directions.

## 1. Introduction

Malignant tumours of the hypopharynx, nasopharynx, oral cavity, oropharynx, and larynx are commonly grouped under the term head and neck squamous-cell carcinoma (HNSCC) [1]. Although these tumours have long been mainly linked to alcohol and tobacco use, human papillomavirus (HPV) infection plays an important role in the development of some HNSCCs, particularly in the oropharynx.

In the last two decades, the incidence of oropharyngeal squamous-cell carcinoma (OPSCC) has increased due to the growing prevalence of HPV infection, which now accounts for approximately 30% of all cases worldwide but with marked geographic differences [2]. In some regions of the world such as North America and northern Europe, most cases of OPSCC are HPV-related [3], probably due to societal changes in sexual practices (i.e., more sexual partners and an increase in oral sex). In many European countries, more than 50% of oropharyngeal cancers are HPV+ [4].

Given the markedly better prognosis for HPV+ OPSCC, HPV infection has important treatment implications [3] as indicated in the most recent edition (eighth) of the American Joint Committee on Cancer (AJCC) staging system, which now defines HPV+ and HPV− OPSCC as separate entities with different characteristics and molecular profiles and distinct outcomes [5]. Although our understanding of HPV+ HNSCC continues to grow as new studies are continually published, it is crucial to keep abreast of the latest evidence.

In this context, the present narrative review provides a comprehensive overview of HPV-driven head and neck cancer, with an emphasis on the current situation in Europe. We review the latest evidence on epidemiology, diagnosis, and treatment, including recent trends towards treatment de-intensification and future directions.

## 2. Epidemiology

HNSCC has long been associated with lifestyle factors, mainly heavy alcohol use, smoking, and chewing tobacco [6]. Most (≈75%) squamous cell carcinomas are caused by alcohol and tobacco use, with the remaining 25% largely attributed to HPV [6]. Most cancers of the head and neck region arise from squamous epithelial cells lining the pharynx, larynx, oral cavity, and/or nasal cavity and these cancers are typically grouped according to their location. HNSCCs are commonly divided into oropharyngeal and non-oropharyngeal carcinomas due to the important role of HPV infection in oropharyngeal cancers [1].

Early studies found that HPV+ OPSCCs were mainly concentrated in younger patients, but more recent data suggest that, in fact, HPV-driven OPSCCs are detected in patients at all ages [4,7,8,9,10,11], with some studies showing that the mean age is comparable between HPV+ and HPV− tumours [12]. Crucially, HPV+ OPSCC has a better prognosis, is more responsive to radiation therapy, and has better overall survival rates than HPV− disease [13,14,15].

In recent years, the incidence of HPV+ OPSCC has increased more rapidly than most other types of cancer in high-income countries, particularly in Europe and the United States [4]. According to Global Cancer Observatory (GLOBOCAN) 2020 data, the incidence of oropharyngeal cancer in Europe was among the highest in the world [16].

### 2.1. Incidence and Prevalence of HNSCC

Figure 1 shows the estimated age-standardised rate (ASR) per 100,000 in 2022 for HNSSC in adults of both sexes in Europe [17]. As that figure shows, the highest rates were observed in Romania, Hungary, and Slovakia, while the lowest rates were observed in Cyprus and Iceland.

As Figure 2 shows [17], the highest mortality rates for HNSCC in Europe in the year 2022 were in Central and Eastern Europe (Moldavia, Romania, Belarus, Hungary) and the lowest rates were in the Nordic countries (Norway, Sweden, Finland) and in Cyprus, Iceland, and Luxembourg.

#### 2.1.1. Oropharynx

According to GLOBOCAN 2022 data [17], an estimated 29,800 new cases of oropharyngeal cancer were diagnosed in 2022 in Europe (ASR = 2.2 per 100,000), with an estimated 13,027 deaths.

HPV positive and negative OPSCC differ in many ways [4,12]. Men make up a larger % of HPV+ OPSCC, accounting for 86.9% of cases versus 76.8% of cases in HPV− disease. Similarly, the mean number of sexual partners is greater in HPV+ disease. Interestingly, despite what was initially thought, HPV+ OPSCC can occur in patients at any age [4,7,8,9,10,11], and some studies suggest that that the median age at diagnosis is similar (59 vs. 60 years) [12].

Stjernstrøm et al. [18] reviewed data from seven studies (2075 patients) conducted in different European countries, finding large differences between countries in terms of the prevalence of HPV+ OPSCC, which ranged from 18% to 65%. The lowest rates were observed in central and southern Europe and the highest in Nordic countries (Sweden and Denmark). These data are largely in line with GLOBOCAN 2022 [16,17].

A recent (2023) systematic review and metanalysis by Fonsêca et al. assessed the global prevalence of HPV in oral cavity and oropharyngeal cancer [19], finding a pooled prevalence of HPV+ disease in 10% and 42%, respectively. The highest prevalence rates for HPV+ OPSCC were observed in northern Europe (Finland and Sweden), a finding that is consistent with aforementioned study by Stjernstrøm et al. [18] and with GLOBOCAN 2020 data [17]. A systematic review by Carlander et al. [20] found an increase in HPV prevalence over time in many European countries, including Germany, the Netherlands, and Italy.

#### 2.1.2. Oral Cancer

The overall prevalence of oral HPV in healthy populations ranges from 1.2% to 11.6%, with high-risk HPV present in 2.2% to 7.2% of individuals and HPV-16 in 0.2% to 2.9% [21]. Although some studies have found that HPV-positive oral cancers have a better prognosis than HPV-negative tumours, other studies have not been able to confirm any association between HPV and longer survival in this patient population [22]. As a result, this question remains unresolved.

#### 2.1.3. Laryngeal Cancer

Although HPV is strongly associated with oropharyngeal cancer, the association with oral and laryngeal cancer is uncertain [23]. However, it is believed that a similar mechanism—p53 inactivation and pRb degradation—drives the proliferation of malignant cells [24].

The epidemiological data on HPV infection in laryngeal cancer vary widely, depending on the region and the study [23,24]. Although HPV infection has been associated with laryngeal cancer, the mechanism of action has not been fully elucidated and the impact of HPV status on survival in these patients remains unclear [23,24]. Evidence to support better survival outcomes in HPV+ laryngeal cancer is lacking. Consequently, the factors that have the greatest impact on survival outcomes in these patients are tumour stage and nicotine exposure.

#### 2.1.4. Cancer of Unknown Primary (CUP)

A recent systematic review by Escobar and colleagues assessed the role of HPV in head and neck cancer of unknown primary (HNCUP) [25]. That review found that HPV is present in a high percentage of cases but with a wide range of reported prevalence rates (from 15.5% to 100%, depending on the study). The role of HPV in HNCUP remains unclear due to conflicting data, with some studies showing an association between HPV positivity and improved survival—such as the study by Axelsson et al., who found that p16 positivity was associated with longer survival and a lower risk of recurrence [26]—and other studies reporting no association [25]. Given the uncertain role of HPV on clinical outcomes in HNCUP, most authors believe that patients diagnosed with HNCUP should all receive the same standard treatment—regardless of HPV status—until more conclusive data become available [27]. Interestingly, the findings of the Head and Neck 5000 study, a large prospective clinical cohort study carried out in the United Kingdom, suggested that most cases of HPV+ HNCUP are likely to be due to HPV-driven OPSCC [28]. However, this finding is controversial and more data will be needed to confirm this hypothesis.

A review by Golusinski et al. [29] on the diagnostic evaluation of cervical node metastases from HNCUP concluded that a complete diagnostic workup consisting of clinical assessment, conventional imaging, FDG positron-emission tomography (PET)/computed tomography (CT) and tonsillectomy/panendoscopy can detect the primary site in more than 50% of cases, which in turn is associated with significantly better survival. Those authors developed a novel, evidence-based protocol designed to identify the primary tumour and determine the disease stage.

### 2.2. The Role of HPV in Carcinogenesis

The molecular and clinical characteristics of tobacco and/or alcohol-induced squamous cell carcinomas differ greatly from HPV-induced OPSCC [5,30]. The carcinogenesis of HPV+ OPSCC involves specific molecular pathways and biological changes characterised by degradation of p53, inactivation of the Rb pathway, and increased p16INK4a (p16) expression. The molecular landscape of HPV-induced OPSCC involves interactions with host cells, particularly through the viral oncoproteins E6 and E7, which target the tumour suppressor proteins (p53 and Rb), leading to uncontrolled cell growth and genomic instability. Other pathways such as PI3K/AKT/mTOR, Notch signalling, and Wnt/β-catenin have also been implicated in the development of HPV+ OPSCC [31,32,33].

The exact mechanisms of HPV infection in the development of HNSCC are not entirely clear [34]. Additionally, the clinical significance of active viral oncogenesis in other non-oropharyngeal locations has not been confirmed. Factors that influence viral susceptibility to tonsillar epithelial crypt is still under research, but it may be due to the fact that HPV virus needs the access to basal cells in squamous epithelium which is much easier in tonsillar crypts. What is more, the microenviroment of tonsillar tissue, which is rich in immune cells, may play a role [35].

Leemans et al., in their article describing the molecular landscape in HNSCC, suggested that productive HPV infections in the oral cavity or larynx rarely change into transforming (i.e., oncogenic) infections. By contrast, transforming infections in the oropharynx typically involve the deregulation of HPV E6 and E7 oncoprotein expression in rapidly dividing cells. In turn, these deregulated oncoproteins inactivate RB and p53, thus supporting progression towards cancer [35].

## 3. Diagnosis

### 3.1. Clinical Presentation

HNSCC mainly affects older adults—particularly men—with a median age at diagnosis of 53 years for HPV+ disease and 66 years for HPV− HNSCC [36]. The signs and symptoms of HNSCC vary widely depending on the primary tumour site and aetiology.

The initial assessment typically involves a physical examination with close study of the head and neck region, including all nodal levels and mucosal surfaces. It is also important to palpate the base of tongue and tonsils. Flexible nasolaryngoscopy is an essential component of the examination [11].

Imaging plays a key role in the diagnostic work-up of patients with suspected HNSCC. In most cases, this consists of CT and magnetic resonance imaging (MRI) followed by a positron-emission tomography (PET)–CT to check for occult metastatic lesions.

### 3.2. Biopsy in HNSCC

The gold standard for the diagnosis of HNSCC is biopsy of the primary tumour followed by determination of HPV status [37]. In some cases, the biopsy can be performed in the clinic with local anaesthesia. If not feasible, then a biopsy must be performed under general anaesthesia. Alternatively, in cases in which the local biopsy was unsuccessful and the patient is not a candidate for open biopsy with general anaesthesia, fine-needle aspiration (FNA) of the cervical lymph nodes may be performed. In these cases, the diagnosis is usually made by a multidisciplinary tumour board based on the cytological results of the tissues obtained via FNA, the clinical presentation, and the results of the imaging studies [37].

### 3.3. Oral Cavity

Cancers of the oral cavity classically present with a persistent oral ulcer. At diagnosis, most oral cancers are late stage, which explains the high mortality rates associated with this type of HNSCC [36,38]. The co-occurrence of risk factors, including tobacco or alcohol use, poor dentition, and use of areca nut and betel quid, should increase diagnostic suspicion. To date, no effective screening strategies have been identified, which is why clinical examination is the primary approach to early detection [36]. The presence of pre-malignant lesions in the oral cavity—leukoplakia, erythroplakia, and submucosal fibrosis—may indicate early-stage disease. However, these often remain undetected and most patients presenting with advanced disease.

While early-stage oral cavity cancers are generally treated with surgery alone, more advanced oral cavity tumours require a multimodal approach consisting of surgical resection followed by adjuvant radiotherapy or chemoradiotherapy (CRT), as appropriate (based on the stage).

### 3.4. Oropharyngeal and Hypopharyngeal Tumours

The signs and symptoms of oro- and hypopharyngeal tumours may include hoarseness, persistent sore throat, a neck mass, dysphagia, odynophagia, or otalgia. Because HPV+ OPSCC is more likely to present with nodal involvement, these patients are also more likely to have a neck mass. Non-specific symptoms (e.g., otalgia) are common, which is why the presence of persistent symptoms is indicative of a need for early referral to specialists [39].

In many cases, primary tumours of the oropharynx do not become symptomatic until the disease is well advanced. However, the presence of non-specific pharyngeal symptoms in patients with a history of tobacco and alcohol use should increase the level of suspicion.

Patients with HPV+ OPSCC typically present with a painless level II neck mass and a primary tumour that is asymptomatic. In these cases, the traditional risk factors for HPV− disease are frequently not present. The two primary risk factors for HPV+ HNSCC are multiple lifetime sexual partners and male sex.

### 3.5. Laryngeal Tumours

Patients with laryngeal cancer often exhibit voice changes or hoarseness, which allows for early diagnosis. However, some individuals may overlook these symptoms and go on to develop dyspnoea and, potentially, obstruction of the airways. Tobacco and alcohol use, which are significant behavioural risk factors, can have a particularly strong impact on the larynx and other HPV-negative sites due to their synergistic effects.

### 3.6. Role and Determination of HPV Status

Given the important differences between HPV+ and HPV− OPSCC in terms of the biological behaviour of the disease, accurate determination of HPV status is essential [5]. HPV-negative tumors frequently exhibit mutations or deletions in the *CDKN2A* gene, which encodes p16, a protein that normally inhibits cyclin-dependent kinases and restricts cell cycle progression. In HPV-positive cancers, p16 is paradoxically overexpressed as a compensatory response to E7-mediated Rb degradation, making it a useful biomarker for distinguishing HPV-driven tumors [40]. Interestingly, p16 only has prognostic value for OPSCC but not tumours of the hypopharynx, larynx, or oral cavity, which is why determination of p16 status is not considered obligatory in non-OPSCC head and neck cancers [41]. Once the primary disease has been identified and HPV status determined, most cases are presented to a multidisciplinary tumour board to select the optimal therapeutic strategy.

Table 1 summarises the clinical role of p16 according to the tumour localisation. As that figure shows, the role of p16 in laryngeal and oral cancer remains unclear. By contrast, it plays an important role in other locations.

In many cancers of the head and neck—particularly OPSCC—the treatment approach may be influenced by HPV status. One of the downstream biological effects of HPV infection is p16 overexpression. The two most common strains—HPV-16 and HPV-18—account for up to 95% of oral cancers and 89% of oropharyngeal cancers [19,42].

Other strains of the virus include HPV-33, HPV-35, and others, but all together it is still under 5% of cases. In other locations, including larynx and oral cancer in Europe, HPV-16 is still the most common one; others, like HPV-33, HPV-45, and HPV-58, are detected at lower frequencies [43].

According to the clinical practice guidelines of the European Society for Radiotherapy and Oncology (ESTRO), the European Head and Neck Society (EHNS), and the European Society for Medical Oncology (ESMO), all newly diagnosed OPSCC patients should undergo p16 immunohistochemistry (IHC) for HPV. The eighth edition of the AJCC also recommends p16 IHC for HPV determination in OPSCC [44].

P16 IHC is a quick and inexpensive assay and widely considered to be a dependable surrogate marker of HPV positivity in the oropharynx [45]. However, due to the high false positive (10–15%) and false negative rates [34], clinical guidelines recommend performing another specific HPV test such as in situ hybridisation (ISH) targeting DNA or RNA to confirm HPV status. In a study by Mehanna et al., the authors assessed nearly 8000 patients with oropharyngeal cancer; 3805 were p16-positive, of whom over 10% were actually HPV negative [34]. This proportion differed significantly worldwide, with the greatest discordance in Spain (around 30%) [34].

Genetic tests are more reliable than p16 IHC, and they are also more expensive and time-consuming, which is why most clinical guidelines recommend p16 IHC as the first-line test. If the test is positive, then further testing is recommended only if HPV status is likely to influence treatment [46].

Determining the HPV status of patients with OPSCC is crucial for guiding treatment, predicting prognosis, and designing clinical trials [1]. HPV status can be assessed using various methods, including the detection of HPV-related genes or the more commonly used p16 marker detected through immunohistochemistry [45,47]. Overexpression of p16 may indirectly indicate the expression of E6 and E7 proteins, leading to cell-cycle upregulation, and serve as a diagnostic tool for prognosis stratification in oropharyngeal cancer patients. High p16 expression levels are associated with a significant survival benefit [1,48]. Another method of estimating the HPV status in HNSCC tissues is detecting HPV DNA using PCR, detecting the mRNA encoding viral oncogenes E6 and E7 by RT-PCR, or using in situ hybridisation targeting viral DNA (DNA ISH) or RNA (RNA ISH) [1]. Ideally, biomarkers should possess high sensitivity and specificity and be cost-effective and suitable for standardised protocols across different laboratories [48]. One of the promising new methods of confirming the HPV status is detecting the circulating HPV tumoural DNA (ctDNA) in patient plasma using ultrasensitive ddPCR (droplet-based digital PCR) [1].

Importantly, there is a need to assess the two-tiered HPV status determination in order to avoid the undertreatment of p16+/HPV− patients [49]. Table 2 summarises the main tests for determination of HPV status and their advantages and disadvantages.

### 3.7. Prognosis and Staging

The two major determinants of prognosis in HNSCC are tumour stage and HPV status. Prior to the publication of the eighth edition of the AJCC manual in 2017, HNSCC was staged according to the TNM system and HPV status was not considered. The eighth edition made several modifications to improve risk discrimination and the predictive accuracy of the system. Probably the most notable change was the new system to stage HPV-positive OPSCC, which greatly improved prognostic differentiation.

Following histopathological confirmation, staging consists of a comprehensive examination of the head and neck regions (especially the oral cavity) with fibreoptic nasolaryngoscopy when appropriate. Other tests include MRI and/or CT scans to determine locoregional involvement and thoracic CT to check for distant metastases. PET–CT is recommended in patients with locally advanced disease and/or nodal involvement.

For any given stage, patients with HPV+ OPSCC have a better prognosis than those with HPV− disease. However, p16 and HPV status frequently conflict in patients with OPSCC and this discordance has prognostic implications. Patients with discordant test results (i.e., p16−/HPV+ or p16+/HPV−) have a significantly worse prognosis than patients with confirmed (p16+/HPV+) OPSCC [5].

Under the recently revised AJCC staging criteria [5], HPV+ patients are now more likely to be classified as early stage, reflecting the better prognosis of HPV-related OPSCC. Table 3 compares the seventh and eighth editions of the staging system for HPV+ OPSCC. As that table shows, patients with HPV+ disease are more likely to be assigned an earlier stage, reflecting better prognosis.

## 4. Treatment

In general, treatment of HNSCC requires a multimodal approach consisting of a combination of surgery, radiotherapy, and chemotherapy in different sequences based on location.

Treatment of oral cavity cancers generally involves surgery alone for early-stage disease or surgery plus adjuvant radiotherapy or CRT for more advanced disease. For cancers of the pharynx or larynx, the primary approach is CRT. However, laryngeal cancers can often be treated with surgery or radiotherapy alone.

Immune-checkpoint inhibitors (e.g., nivolumab, pembrolizumab) are used to treat metastatic or recurrent HNSCC, with pembrolizumab being the main treatment for nonresectable tumours [36]. Cetuximab, a radiosensitiser, is often used for the treatment of metastatic or recurrent HNSCC, mainly in patients who are not eligible for cisplatin.

Definitive CRT is generally indicated in patients with advanced disease (≥stage T3) with nodal involvement of the neck or to preserve function. The standard chemotherapy regimen is cisplatin (100 mg/m^2^) administered every 3 weeks. Although cetuximab can be used as alternative to cisplatin as a radiosensitiser, survival outcomes are lower in HPV+ OPSCC patients who receive cetuximab [50]. Although CRT has been proven to improve survival, its association with important adverse effects explains the growing interest in treatment de-escalation strategies.

### 4.1. Role of Minimally Invasive Surgery

The significant treatment-related morbidity associated with conventional approaches in HNSCC—surgery plus adjuvant CRT—has stimulated interest in treatment de-intensification, mainly to reduce the long-term toxicity of CRT. While most de-intensification efforts have focused on reducing chemotherapy and/or radiotherapy, there has been a growing interest in minimally invasive surgery to improve functional outcomes with less morbidity [36,51].

In recent years, major advances have been made in transoral robotic surgery (TORS) and transoral laser microsurgery (TLM). In the context of the severe toxicities associated with CRT, this has prompted many researchers and clinicians to evaluate whether transoral surgery could help to avoid or at least reduce the need for radiotherapy and/or chemotherapy [52]. Upfront surgery offers the advantage of histopathologic staging and several authors have reported better quality of life in patients treated with primary surgery [53,54,55]. Other studies have shown that transoral robotic surgery and TLM can achieve comparable cure rates to open surgery when used in conjunction with postoperative, risk-adapted radiotherapy [56]. Peng et al. recently compared survival outcomes in patients with OPSCC treated with upfront surgery versus definitive radiotherapy, finding that upfront surgery resulted in better mortality outcomes, irrespective of HPV status [56]. Several recent retrospective studies have also found that primary transoral surgery (TOS) improves functional outcomes [51,57]. However, these findings will need to be confirmed in prospective clinical trials.

### 4.2. Immunotherapy

In recent years, the development of anti-PD1 checkpoint inhibitors has improved outcomes in patients with advanced-stage HNSCC, regardless of HPV status. A pooled analysis of prospective clinical trials conducted to evaluate the role of PD-1/PD-L1 inhibitors showed that overall survival (OS) and objective response rates (ORR) were better in HPV+ patients than in HPV− patients [58].

### 4.3. Treatment De-Intensification

Patients with HPV+ OPSCC tend to present with fewer comorbidities and other risk factors and have a better prognosis and longer life expectancy than their HPV− counterparts [30,59]. However, treatment-related morbidity is a major concern, which is why there has been a major effort in recent years to reduce the morbidity associated with highly aggressive treatments, an approach widely known as treatment de-intensification. Conceivably, it may be possible to achieve similar oncological outcomes with less toxicity [44].

Patients with HPV+ OPSCC tend to be younger and have fewer comorbidities than their HPV− counterparts, with a better prognosis. Not surprisingly, this situation has prompted efforts to de-intensify treatment to minimise the severe morbidity associated with cisplatin and radiation [60]. However, the first three randomised trials carried out to evaluate treatment de-escalation found that omitting or replacing cisplatin had a clear negative impact on survival [60]. As Mehanna et al. [61] observed, more data are needed from phase II trials to better understand the feasibility of de-escalation before we can conduct phase III trials. In other words, more robust evidence is needed before we consider implementing these strategies in the clinic.

Numerous trials have been performed or are currently underway to identify the optimal de-intensification strategy in HPV-related OPSCC [15]. Sung et al. [62] recently reviewed the prospective de-intensification trials. Those trials evaluated a variety of different de-escalation strategies, including reduction in the radiation dose in definitive treatment, substitution of platinum-based chemotherapy with cetuximab, personalised dose prescriptions based on response to induction chemotherapy, and decrease in adjuvant radiation doses after transoral surgery. Kang et al. recently (2023) reviewed the prospective HPV-positive OPSCC de-escalation trials, finding that chemotherapy attenuation negatively impacts clinical outcomes but does not reduce toxicity [15]. Although there is some evidence to support a benefit for de-escalating adjuvant treatment after TORS, the treatment indications remain poorly defined. Although some studies have reported promising results with radiation de-intensification strategies, robust data are lacking [15]. Based on their findings, Kang and colleagues concluded that tumour stage and HPV status may not be enough to guide the de-intensification strategy. Rather, they suggest that de-intensification strategies may need to focus on assessing intra-treatment response and real-time surveillance.

Molteni et al. recently reviewed the role of TORS for de-escalation in HPV-related OPSCC [63]. The findings of that comprehensive review underscored the potential role for TORS to reduce the need for adjuvant therapy and thus the side effects of those treatments. Several notable trials are currently underway in Europe to evaluate the value of TORS for de-intensification, including the PATHOS trial [64] and the EORTC Best Of trial (NCT02984410) [65]. In the EORTC trial, patients with T1–T2N0–N1 OPSCC (any HPV status) or supraglottic larynx cancer or T1N0 hypopharynx cancer will be randomised to receive TORS or standard radiotherapy. The PATHOS trial is an ongoing phase II/III risk-stratified trial involving reduced-intensity adjuvant treatment to assess the impact of risk-adapted adjuvant treatment on functional outcomes and survival. That trial seeks to determine whether de-intensification of adjuvant therapy can reduce chronic dysphagia without negatively influencing clinical results. O’Hara et al. recently compared patients included in the PATHOS trial who underwent TLM or TORS, finding that TORS was associated with significantly higher rates of nasogastric tube NGT use, worse swallowing scores (H&N35), and worse MDADI (MD Anderson Dysphagia Inventory) scores at 4 weeks post-surgery [66].

As Molteni and colleagues conclude in their review, TORS is a feasible upfront treatment option for well-selected patients with HPV+ OPSCC. Nevertheless, patient selection is essential to select the most suitable candidates for TORS-based de-intensification approaches.

Despite the growing interest in treatment de-intensification, as evidenced by the large number ongoing trials, the NCCN (National Comprehensive Cancer Network) and other professional associations have stated that this approach should be limited to clinical trials until more evidence from phase III randomised controlled trials is available [44].

### 4.4. Future Directions

Numerous trials are currently underway to assess the potential for treatment de-escalation, as evidenced by the large number of trials registered at Clinicaltrials.gov. A search of that database (24 May 2024) using the key words “head OR neck AND cancer” and “de-escalation” yields a total of 157 studies.

In addition, intense research efforts are underway to identify targeted therapies and novel immunotherapies [67]. Since many head and neck cancers, particularly HPV-positive ones, are only detected at an advanced stage, there is a clear, urgent need to identify biomarkers for early detection [31]. While recent systematic reviews and meta-analyses have identified several promising biomarkers that could be used for early detection [68], more data are needed to determine the most reliable of these potential biomarkers.

We also need to further improve our understanding of the molecular basis of these cancers and to focus more on early detection and precision care, which should help to improve patient outcomes [4]. In addition, further research is needed to elucidate the process by which oral HPV infection progresses to cancer [31,32,33].

Prevention is another area of need, particularly in the context of the ever-increasing burden of HPV+ OPSCC. In this regard, an increase in HPV vaccination rates would reduce infection rates more cancer screening would likely increase detection at earlier stages, thus allowing for early interventions to reduce morbidity and mortality in these patients [31]. In terms of vaccination in Europe, most countries integrated HPV vaccination into their national immunisation programs, but only 14 countries included boys. Vaccination coverage varies significantly with high percentages in Belgium and Iceland (around 90%), a moderate percentage in Norway and Finland (around 70%), and low percentages in France and Poland (under 20%). Expanding vaccination programs to include all genders and enhancing public education are essential steps toward mitigating the impact of HPV-positive HNCSCC in Europe [69].

## 5. Conclusions

The present review provides an overview of the current status of HPV-driven HNSCC in Europe. As this review demonstrates, the incidence of HPV+ HNSCC is still increasing, posing a significant challenge to healthcare systems worldwide, but particularly in Europe and other high-income regions.

Although it is clear that HPV positivity plays a key role in the development of HNSCC, the biology of HPV-dependent cancer has not yet been fully elucidated and the biological behaviour of these tumours differs depending on the location.

Numerous studies have been performed or are currently underway to determine whether treatment de-escalation can be performed safely without compromising oncological outcomes. However, more robust data are needed before treatment de-escalation can be offered to patients outside of clinical trials. However, there are intense efforts underway worldwide to identify reliable biomarkers for early detection, which would have a major positive impact on treatment outcomes.

## Figures and Tables

**Figure 1 viruses-17-00662-f001:**
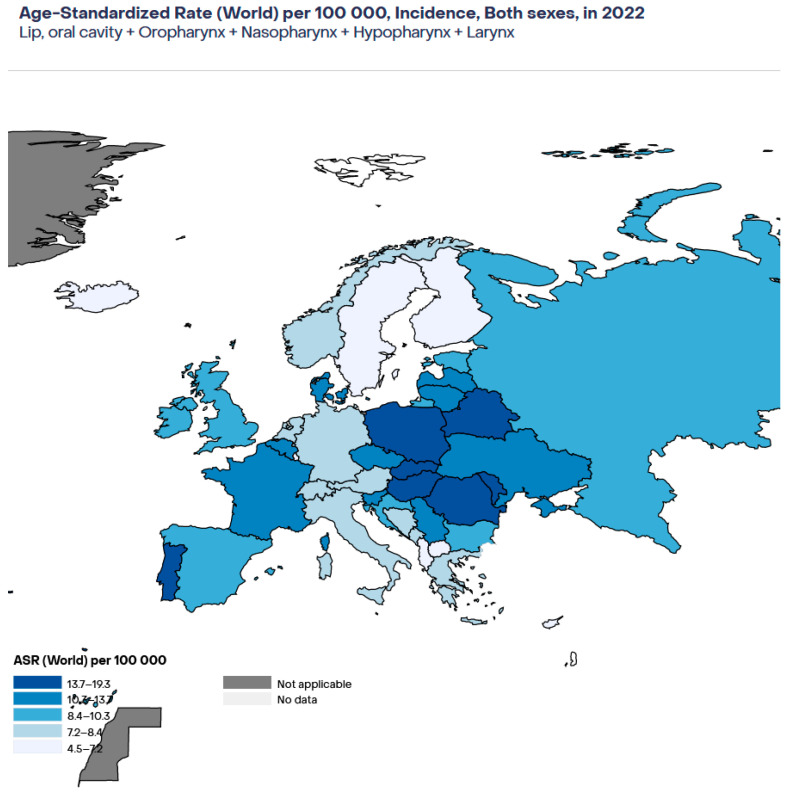
Estimated age-standardised incidence of HNSCC in Europe in 2022. Age-standardised rate.

**Figure 2 viruses-17-00662-f002:**
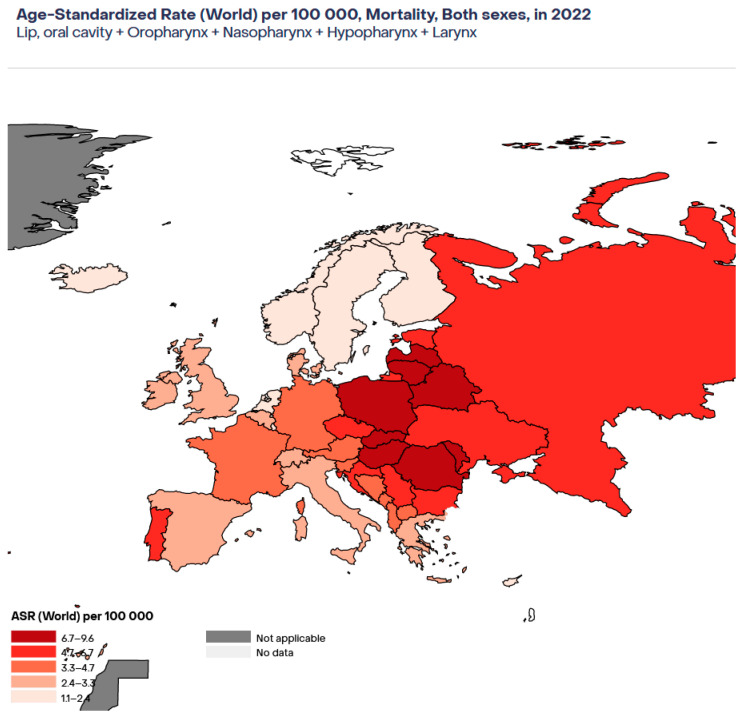
Estimated age-standardised mortality rates for HNSCC in Europe in 2022.

**Table 1 viruses-17-00662-t001:** Clinical role of p16 according to tumour location.

Cervix	p16 Overexpression as a Marker for the Oncologic Progression
Oropharynx	p16+ patients have much better prognosis
CUP	Suggestion for primary tumour location
Larynx	Role unknown
Oral	Role unknown

**Table 2 viruses-17-00662-t002:** Determination of HPV status: available assays and their advantages and disadvantages.

Detection Technique	Advantages	Disadvantages
HPV PCR	High sensitivity Data on HPV genotype FFPE manageable Easy and inexpensive	No data on viral transcription High risk of contamination
E6/E7 mRNA RT-PCR	High sensitivity and specificity Detects active viral infection Gold standard for research	Slow processing Non-FFPE manageable (fresh or frozen tissue only) RNA fragile
E6/E7 mRNA in situ hybridisation	High specificity and good sensitivity In situ detection FFPE manageable	RNA degradation over time Expensive
HPV DNA in situ hybridisation	In situ detection High specificity FFPE manageable	Reduced sensitivity (needs a minimum DNA copy number)
P16 immunohistochemistry	High sensitivity Inexpensive FFPE manageable	Moderate specificity Surrogate marker of HPV infection
Serology for antibodies against E6 protein	Present in more than 90% of patients with OPSCC related to HPV16 Easy to set up	More clinical data needed to support.
HPV circulating tumoral DNA by ddPCR	Correlation with clinical outcome Easy detection of recurrences in posttreatment monitoring High sensitivity and specificity Inexpensive	Not validated in large cohorts

Abbreviations: RT-PCR, real-time polymerase chain reaction; ddPCR, droplet digital PCR; HPV, human papilloma virus; FFPE, formalin-fixed paraffin-embedded; OPSCC, oropharyngeal squamous-cell carcinoma.

**Table 3 viruses-17-00662-t003:** Comparison of AJCC 7th and 8th editions for staging HPV+ oropharyngeal cancer. Shaded areas indicate changes.

**AJCC 7th Edition, OPSCC**				
	**N0**	**N1**	**N2a,b,c**	**N3a,b**
T1	I	III	IVa	IVb
T2	II	III	IVa	IVb
T3	III	III	IVa	IVb
T4a	Iva	IVa	IVa	IVb
T4b	IVb	IVb	IVa	IVb
**AJCC 8th Edition, OPSCC**				
	**N0**	**N1**	**N2**	**N3**
T1	I	I	II	III
T2	I	I	II	III
T3	II	II	II	III
T4	III	III	III	III

Abbreviations: AJCC, American Joint Committee on Cancer; OPSCC, oropharyngeal squamous-cell carcinoma.

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
