# Peer review of "HPV-Driven Head and Neck Cancer: The European Perspective"

_viruses, 2025, doi:10.3390/v17050662_

Round 1

Reviewer 1 Report

Comments and Suggestions for Authors

this manuscript offers little that is new

it is a review   

Author Response

Dear Reviewer,

first of all I would like to thank you for your time and effort spent while reading our manuscript. We respect your oppinion, nevertheless there is a lack of European insight of HPV in head and neck cancer. 

Reviewer 2 Report

Comments and Suggestions for Authors

This paper effectively reviews the diagnosis, management and epidemiology of HPV associated head and neck cancers in Europe.  The clinical aspects and management of HPV+ve and -ve cancers of the head and neck are comprehensively reviewed, the references are relevant and contemporary.  I have some criticisms, the epidemiology is covered in a rather perfunctory manner and the discussion of risk factors is inadequate.   Page 8 lines 208-210 should be accompanied by supporting references particularly the noting the original study from Gillison

Gillison ML. Current topics in the epidemiology of oral cavity and oropharyngeal cancers. Head Neck. 2007;29(8):779-792.

The authors are not molecular virologists but they must give a more detailed comment on why p16 expression is a marker of HPV gene expression plus some comment on the importance of accurate detection of HPV.  The biology of the tonsillar epithelium and its susceptibility to HPV infection and subsequent cancerous change is ignored.

Overall this is a good review for head and neck oncologists and surgeons but is flawed in the basic science.

Author Response

Dear Reviewer,

first of all I would like to thank you for your thorough revision. Going into details, our manuscript was revised as follows:

Comment 1:   Page 8 lines 208-210 should be accompanied by supporting references particularly the noting the original study from Gillison

Response 1: The citation has been added to 3.6 section (Role and determination of HPV status)

Comment 2: The authors are not molecular virologists but they must give a more detailed comment on why p16 expression is a marker of HPV gene expression plus some comment on the importance of accurate detection of HPV.

Response 2: These sentences have been added to 3.6 section: "...HPV-negative tumors frequently exhibit mutations or deletions in the CDKN2A gene, which encodes p16, a protein that normally inhibits cyclin-dependent kinases and restricts cell cycle progression. In HPV-positive cancers, p16 is paradoxically overexpressed as a compensatory response to E7-mediated Rb degradation, making it a useful biomarker for distinguishing HPV-driven tumors (67)....In a study by Mehanna et al the authors assessed nearly 8000 patients with oropharyngeal cancer. 3805 were p16 positive, of whom over 10% were actually HPV negative. This proportion differed significantly worldwide, with greatest discordance in Spain (around 30%) (34)."

Comment 3: The biology of the tonsillar epithelium and its susceptibility to HPV infection and subsequent cancerous change is ignored.

Response 3: This paragraph has been added to 1.2 section: 

"The exact mechanisms of HPV infection in the development of HNSCC are not entirely clear(34). Additionally the clinical significance of active viral oncogenesis in other non-oropharyngeal locations have not been confirmed. Factors that influence viral susceptibility to tonsillar epithelial crypt is still under research but it may be due to the fact that HPV virus needs the access to basal cells in squamous epithelium which is much easier in tonsillar crypts. What is more, the microenviroment of tonsillar tissue which is rich in immune cells may play a role (65)"

Reviewer 3 Report

Comments and Suggestions for Authors

This narrative review focuses on the role of human papillomavirus in the etiology, epidemiology, diagnosis, and treatment of head and neck squamous cell carcinoma, particularly oropharyngeal squamous cell carcinoma. The review highlights key updates, such as the reclassification of HPV+ and HPV- OPSCC as distinct clinical entities in the AJCC 8th edition staging manual. Emphasis is placed on the European context, including regional variations in HPV prevalence and emerging treatment strategies such as de-intensification for HPV-positive patients. The reviewer has the following suggestion to improve the manuscript.

  1. The authors must include a detailed narration on the scenario of HPV vaccination and its efficacy in reducing HPV+ HNSCC in European region.
  2. What do the authors mean in line 223, “which is shy determination of p16 status is not considered obligatory in non-OPSCC head and neck cancers”? Please clarify.
  3. The authors must include a paragraph emphasizing the most common HPV subtypes identified in European region other than HPV-16 and HPV-18 and their prevalence in context of oral, oropharyngeal or larynx. 

Author Response

Dear Reviewer,

first of all thank your for your time spent reviewing our manuscript and your valuable comment. Going into details:

Comment 1: The authors must include a detailed narration on the scenario of HPV vaccination and its efficacy in reducing HPV+ HNSCC in European region

Response 1: Added to 4.4 section: 

In terms of vaccination in Europe most countries integrated HPV vaccination into their national immunization programs, but only 14 countries included boys. Vaccination coverage varies significantly with high percentage in Belgium and Iceland (around 90%) moderate in Norway and Finland (around 70% and low in France and Poland (under 20%) Expanding vaccination programs to include all genders and enhance public education are essential steps toward mitigating the impact of HPV positive HNCSCC in Europe (68).

Comment 2: What do the authors mean in line 223, “which is shy determination of p16 status is not considered obligatory in non-OPSCC head and neck cancers”? Please clarify

Response 2: It was meant to be: this is why... It is not considered obligatory because it is not a prognostic factor in non-OPSCC cancers"

Comment 3: The authors must include a paragraph emphasizing the most common HPV subtypes identified in European region other than HPV-16 and HPV-18 and their prevalence in context of oral, oropharyngeal or larynx.

Response 3: Twe his was added to 3.6 section: 

Other strains of the virus include HPV-33, HPV-35 and other but all together it’s still under 5% of cases. In other locations, including larynx and oral cancer in Europe HPV-16 is still the most common one, others like HPV-33, HPV-45 and HPV-58 are detected at lower frequencies (69)

Round 2

Reviewer 3 Report

Comments and Suggestions for Authors

The authors have carefully revised the manuscript making the review comprehensive and acceptable for publication.